# Jolkinolide B inhibits the progression of hepatocellular carcinoma by regulating Musashi-2 protein

**Tianchun Wu**[1,2]*, **Han Yang**[1,2], **Jinjin Li**[1,2], **Hongbo Fang**[1,2], **Xiaoyi Shi**[1,2], **Jie Li**[1,2], **Liushun Feng**[1,2]

1 Division of Hepatobiliary and Pancreatic Surgery, The First Affiliated Hospital of Zhengzhou University, Zhengzhou, Henan Province, China, 2 Division of Hepatobiliary and Pancreatic Surgery, Henan Key Laboratory of Digestive Organ Transplanation, The First Affiliated Hospital of Zhengzhou University, Zhengzhou, Henan Province, China

* doctor_wtc518@zju.edu.cn

## Abstract

Hepatocellular carcinoma (HCC) is one of the leading causes of cancer-related deaths. However, the HCC treatment is still challenging. Herein, we aimed to reveal the anti-tumor effect of Jolkinolide B in HCC cell lines Huh-7 and SK-Hep-1. The results showed that Jolkinolide B inhibited the migration, invasion, and epithelial-to-mesenchymal transition(EMT) of HCC cells. In addition, Jolkinolide B induced HCC cell apoptosis by upregulating Bax and downregulating BCL-2 expressions. Furthermore, we demonstrated that Jolkinolide B inactivated the β-catenin signaling and reduced Musashi-2 expression. Finally, we revealed that Musashi-2 overexpression reversed the Jolkinolide B-induced anti-HCC effect. Overall, we proved that Jolkinolide B is a potential candidate for treating HCC.

## Introduction

Hepatocellular carcinoma (HCC) is one of the most common malignant tumors [1]. Although the survival rate of patients with HCC has improved, HCC is the fifth leading causes of cancer-related deaths worldwide, and effective therapeutic approaches for HCC in clinical treatment are still lacking [2, 3]. Therefore, developing efficient pharmaceutical agents against HCC is urgently necessary.

Musashi-2 (MSI2), a member of the MSI family, is a conserved RNA-binding protein [4, 5]. MSI2 is an important regulation factor of sperm and embryo formation. Currently, numerous studies have shown that MSI2 plays a crucial role in mediating cancer stemness, migration, invasion, proliferation, and EMT. In addition, MSI2 can be a prognostic factor for tumors [6–8]. For instance, MSI2 has been identified as a prognostic biomarker for gastric, cervical, and non-small cell lung cancers [9–11]. Moreover, MSI2 promotes bladder cancer migration and invasion by activating the JAK2/STAT3 signaling pathway [6].

Jolkinolide B, extracted from the traditional Chinese herb *Euphorbia fischeriana Steud.*, is a bioactive diterpenoid with anti-tumor effects against several tumors [12]. For example, Jolkinolide B induces gastric cancer cell apoptosis and cell cycle *in vitro*, and inhibits tumor growth *in vivo*. In addition, Jolkinolide B induces MCF-7 cell apoptosis by inactivating the PI3K/Akt/

**Funding:** Medical scientific and technological research joint construction projects of Henan province,2020. NO.LHGJ20200361.

**Competing interests:** The authors have declared that no competing interests exist.

mTOR signaling pathway [13, 14]. Jolkinolide B inhibited the breast cancer MDA-MB-231 cell line owing to its anti-metastatic effects [15]. However, the effects of Jolkinolide B on HCC remains unclear. In this study, we explored the biological effects and mechanisms of action of Jolkinolide B on HCC. This study provides a novel therapeutic approach for HCC.

## Materials and methods

### Cell lines and reagents

Human L-02 cells and HCC Huh-7 and SK-Hep-1 cells were maintained in Dulbecco's Modified Eagle Medium (DMEM) (Invitrogen, Shanghai, China) supplemented with 10% heat-inactivated fetal bovine serum, 100 U/mL penicillin, and 100 mg/mL streptomycin sulfate. The cells were incubated at 37 ºC in a 5% $CO_2$ incubator. Jolkinolide B was purchased from Med-ChemExpress and dissolved in dimethyl sulfoxide (DMSO).

### Cell viability assay

The antiproliferative effect of Jolkinolide B was evaluated in L-02, Huh-7 and SK-Hep-1 cell lines. Briefly, $2 \times 10^3$ cells were seeded in 96-well plates and cultured overnight. In the following day, cells were treated with different concentrations of Jolkinolide B (0, 5, 10, 25, 50, or 100 μM) for 48 h. Finally, 10 μL Cell Counting Kit-8 (CCK-8) solution (Dojindo Molecular Technologies, Japan) was added to each well. The cells were then cultured for 2 h. The optical density was measured at 450 nm using a microplate reader (Thermo Fisher Scientific Inc., Waltham, MA, USA).

### Lentivirus-mediated MSI2 overexpression

Huh-7 and SK-Hep-1 cells were seeded in six-well plates and incubated overnight at 60% confluence. Lentivirus-mediated MSI2 plasmids were added to HCC cells and incubated for 48 h. Then, the total protein and RNA were extracted from the cells for subsequent experiments, as described in the following sections.

### Western blotting

Cells were lysed using radioimmunoprecipitation assay buffer (Thermo Fisher Scientific Inc.) containing phenylmethylsulfonyl fluoride and phosphatase inhibitors. The proteins were quantified using bicinchoninic acid protein assay. Proteins were then separated using sodium dodecyl sulfate-polyacrylamide gel electrophoresis, and transferred to polyvinylidene difluoride membranes. Membranes were blocked with 5% nonfat milk at room temperature (20–25 ºC) for 1 h and incubated with primary antibodies at 4 ºC overnight. After incubation, the membranes were washed thrice with $1 \times$ tris-buffered saline with 0.1% Tween$^®$ 20 for 10 min and incubated with secondary antibodies for 1 h at room temperature. Antibodies against MSI2 (ab76148), Bax (ab32503), and BCL-2 (ab32124) were purchased from Abcam (Cambridge, UK), and antibodies against E-cadherin (#14472), vimentin (#5741), GAPDH (#97166), p53 (#9282) and β-catenin (#8480) were obtained from Cell Signaling Technology, Inc., (Danvers, MA, USA).

### Flow cytometry analysis

Jolkinolide B-induced HCC cell apoptosis was evaluated using an Annexin V-FITC Apoptosis Detection Kit (Beyotime, China). Briefly, Huh-7 and SK-Hep-1 cells were cultured in six-well plates and treated with Jolkinolide B. Huh-7 and SK-Hep-1 ($1 \times 10^6$ cells) cell suspension were collected, washed three times with pre-cold phosphate-buffered saline buffer, and stained with

5 μL Annexin V-FITC and 4 μL propidium iodide in the dark for 30 min. The cell apoptosis was analyzed using a Flow Cytometer (BD Biosciences, San Jose, CA, USA). Finally, the HCC apoptotic cell rates were calculated.

### Real-time quantitative polymerase chain reaction

Total RNA was extracted using a TRIzol kit (Invitrogen, Thermo Fisher Scientific, Inc.). The RNA concentration was measured using a NanoDrop spectrophotometer (Thermo Fisher Scientific). The RNA was transcribed into cDNA using the PrimeScript II 1st Strand cDNA Synthesis Kit (Takara, China). The SYBR Green PCR Master Mix method was used to amplify the target genes. The $2^{-\Delta\Delta Ct}$ method was used to analyze the relative expression of the target genes. GAPDH was used as an internal control. The primers used in this study were: MSI2, 5'-ATCCCACTACGAAACGCTCC-3' (forward) and 5'-GGGGTCAATCGTCTTGGAATC-3' (reverse); matrix metalloproteinase-7 (MMP-7), 5'-GAGTGAGCTACAGTGGGAACA-3' (forward) and 5'- CTATGACGCGGGAGTTTAACAT-3' (reverse); c-Myc, 5'-GGCTCCTGGCAAAAGGTCA-3' (forward) and 5'-CTGCGTAGTTGTGCTGATGT-3' (reverse); and GAPDH, 5'-CTGGGCTACACTGAGCACC-3' (forward) and 5'-AAGTGGTCGTTGAGGGCAATG-3' (reverse).

### Wound healing assay

Huh-7 or SK-Hep-1 cells were seeded in six-well plates overnight and scratched with a 100-μL pipette tip. The scratched wound observed under an inverted microscope, and images were collected. After incubation with DMSO or Jolkinolide B at a concentration of 10 μM for 48 h, the migrated area was photographed.

### Transwell assay

Huh-7 or SK-Hep-1 cells were seeded into the upper chamber of transwell plates in 500 μL DMEM containing DMSO or Jolkinolide B. The lower chamber contained DMEM supplemented with 20% fetal bovine serum. After incubating for 48 h, the cells in the lower chamber were fixed with paraformaldehyde for 1 h and stained with 0.1% crystal violet. Then, the cells were counted under an optic microscope.

### Statistical analysis

Cell experiments were repeated in triplicate. Data were expressed as mean values ± standard deviation. One-way analysis of variance was performed for multiple comparisons among different groups. Data processing and analysis were performed using GraphPad Prism software (La Jolla, CA, USA). A $p$-value $<0.05$ was considered statistically significant, and $^{*}p<0.05$, $^{**}p<0.01$, $^{***}p<0.001$, and $^{****}p<0.0001$.

## Results

### Jolkinolide B reduces HCC cell lines viability

To identify the anti-tumor effects of Jolkinolide B on HCC cells, Huh-7 and SK-Hep-1 cells were treated with different concentrations (0, 5, 10, 25, 50, or 100 μM) of Jolkinolide B for 48 h and the cell viability was assessed using CCK-8 assay. IC50 for Huh-7 and SK-Hep-1 cells were 14.09 μM and 11.17 μM, respectively. The results showed that Jolkinolide B decreased HCC cell viability in a dose-dependent manner. Furthermore, we found that Jolkinolide B did not cause cytotoxicity to L-02 cells (Fig 1). Together, these results indicate that Jolkinolide B inhibits hepatocellular carcinoma cells.

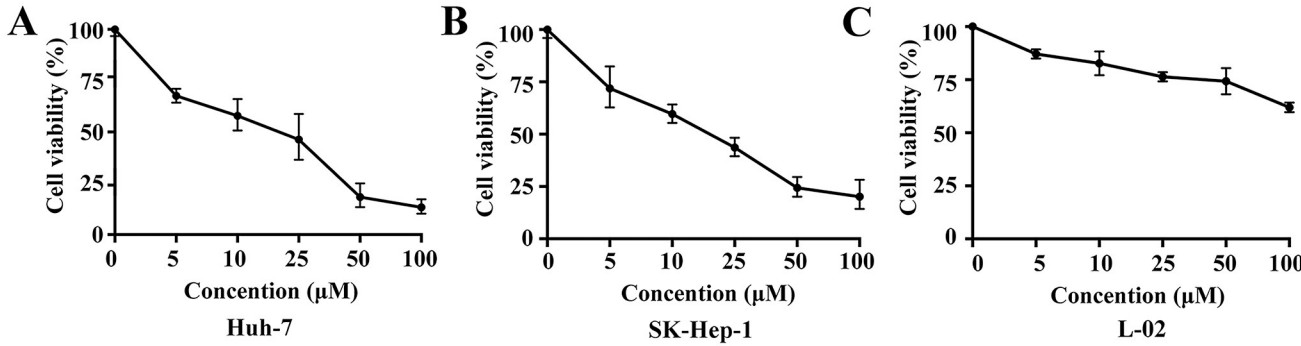

**Fig 1. Jolkinolide B inhibits HCC cells proliferation.** Cell viability assays of Huh-7 (A), SK-Hep-1 (B) and L-02 cells (C) were performed using CCK-8 method after treatment with different concentrations (0, 5, 10, 25, 50, or 100 μM) of Jolkinolide B for 48 h.

## Jolkinolide B inhibits HCC cell migration and promotes HCC cell apoptosis

To investigate the effects of Jolkinolide B on HCC cell migration, invasion, and EMT, we performed wound healing assay using Huh-7 and SK-Hep-1 cells. Transwell assay was performed to estimate the anti-tumor effects of Jolkinolide B on the migration and invasion of HCC cells. The results showed that Jolkinolide B inhibited HCC cell migration and invasion (Fig 2A and 2B). Furthermore, we performed western blotting to estimate the effect of Jolkinolide B on EMT. The results showed that Jolkinolide B upregulated the EMT marker E-cadherin and downregulated vimentin protein expression (Fig 2C). Apoptosis plays an important role in traditional Chinese herbal-induced anti-tumor effect. Therefore, we investigated the effect of Jolkinolide B on HCC cell apoptosis using western blotting to evaluate Bax and BCL-2 protein expressions. The results showed that Jolkinolide B increased Bax protein expression, whereas it decreased BCL-2 protein levels (Fig 2D). Besides, we applied flow cytometry to analyze the apoptosis rates of HCC cells induced by Jolkinolide B, the results showed that Jolkinolide B could induce HCC cells apoptosis (Fig 2E). Collectively, Jolkinolide B inhibited HCC cell migration, invasion, and EMT, and induced cell apoptosis.

## Jolkinolide B inactivates β-catenin signaling pathway

The activation of the β-catenin signaling pathway promotes tumor progression. Therefore, the inactivation of the β-catenin signaling pathway is an effective method for inhibiting tumor progression. To investigate the effect of Jolkinolide B on β-catenin signaling pathway, Huh-7 (Fig 3A) and SK-Hep-1 (Fig 3B) cells were treated with different concentrations of Jolkinolide B (0, 5, 10, 25, 50, or 100 μM). The results showed that Jolkinolide B downregulated β-catenin protein expression in a dose-dependent manner. In addition, the results showed that the downstream genes of β-catenin signaling *MMP-7* and *C-MYC* were also downregulated by Jolkinolide B treatment (Fig 3C and 3D).

## Jolkinolide B downregulates MSI2 expression

Previous studies have reported that MSI2 is an important marker of tumor progression and is vital for tumor metastasis, proliferation, and cell cycle in different types of tumors, including HCC. To investigate whether Jolkinolide B can induce the expression of MSI2, Huh-7 and SK-Hep-1 cells were treated with different concentrations of Jolkinolide B (0, 5, 10, 25, 50, or 100 μM). The results showed that Jolkinolide B downregulated the MSI2 protein expression in

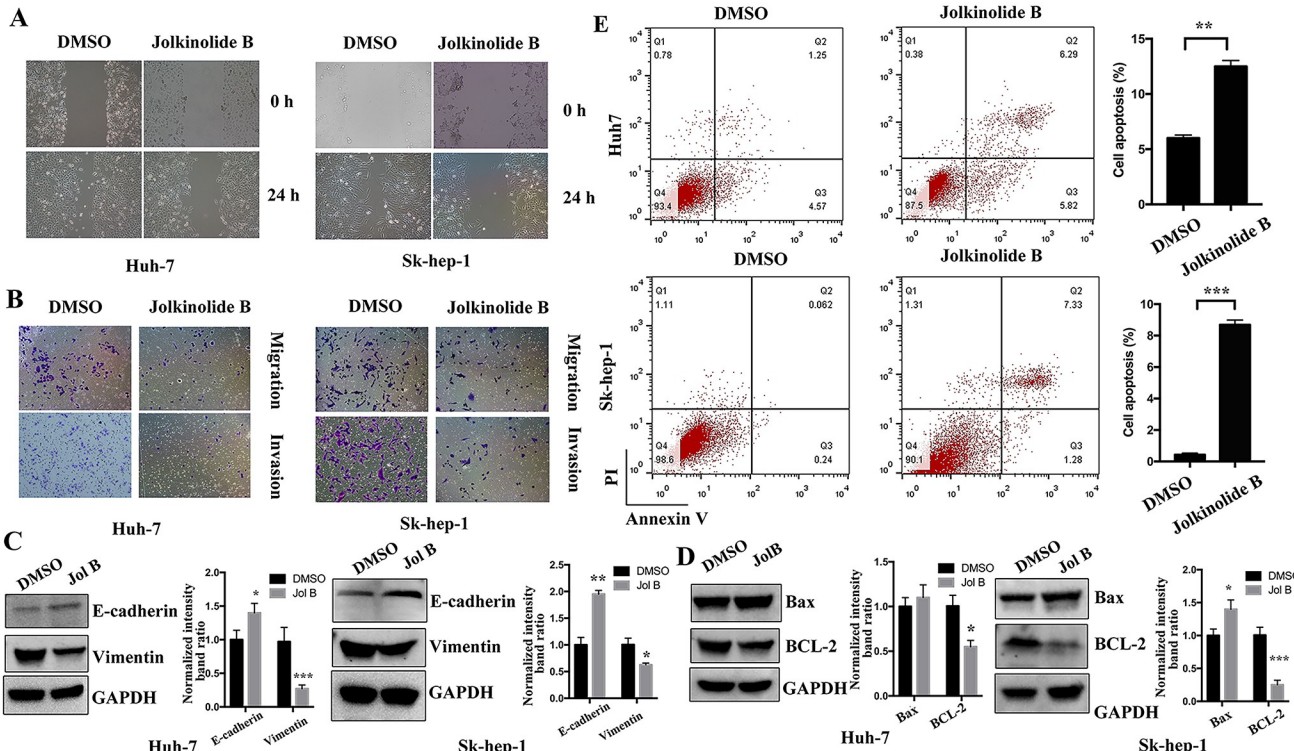

**Fig 2. Jolkinolide B inhibits HCC cell lines migration, invasion and promotes HCC cell apoptosis.** (A) Huh-7 and SK-Hep-1 cells were seeded, scratched, and then treated with Jolkinolide B at a concentration of 10 μM for 48 h. The wound healing assays were performed to assess the migration abilities of Huh-7 and SK-Hep-1 cells. (B) Huh-7 and SK-Hep-1 cells were seeded in the upper transwell chamber and treated with DMSO or Jolkinolide B of 10 μM for 48 h to evaluate the migration and invasion abilities of HCC cells. (C) Huh-7 and SK-Hep-1 cells were treated with DMSO or Jolkinolide B at a concentration of 10 μM for 48 h; then the protein expression of E-cadherin and vimentin was analyzed using western blotting. GAPDH was used as an internal control. (D) Western blotting was used to assess Bax and BCL-2 protein expressions. GAPDH was used as an internal control. The relative protein intensities were analyzed. (E) Huh-7 and SK-Hep-1 cells were treated with DMSO or Jolkinolide B, the cells apoptosis was analyzed by flow cytometry and apoptosis rates were calculated. *$p < 0.05$, **$p < 0.01$ and ***$p < 0.001$.

Huh-7 (Fig 4A) and SK-Hep-1 (Fig 4B) cells. p53 shows as an important cancer suppressor, we detected whether Jolkinolide B induced p53 expression, western blots showed that Jolkinolide B could upregulate p53 expression (Fig 4C and 4D).

## MSI2 overexpression promotes β-catenin signaling

To identify the role of MSI2 on the regulation of β-catenin signaling, this study overexpressed MSI2 expression in both Huh-7 and SK-Hep-1 cells (Fig 5A and 5B). β-catenin was detected after MSI2 overexpression (Fig 5C). These results indicated that MSI2 overexpression promoted β-catenin expression.

## MSI2 overexpression reverses Jolkinolide B-induced inhibition of HCC cells

The above results showed that Jolkinolide B inhibited HCC cell migration, induced apoptosis, and downregulated MSI2 expression. To further investigate the effect of MSI2 in Jolkinolide B-induced HCC progression, Huh-7 and SK-Hep-1 cells were treated with DMSO, Jolkinolide B, or Jolkinolide B and MSI2 plasmids. The results showed that MSI2 reversed Jolkinolide B-induced upregulation of Bax and downregulation of BCL-2 (Fig 6A). Therefore, we further

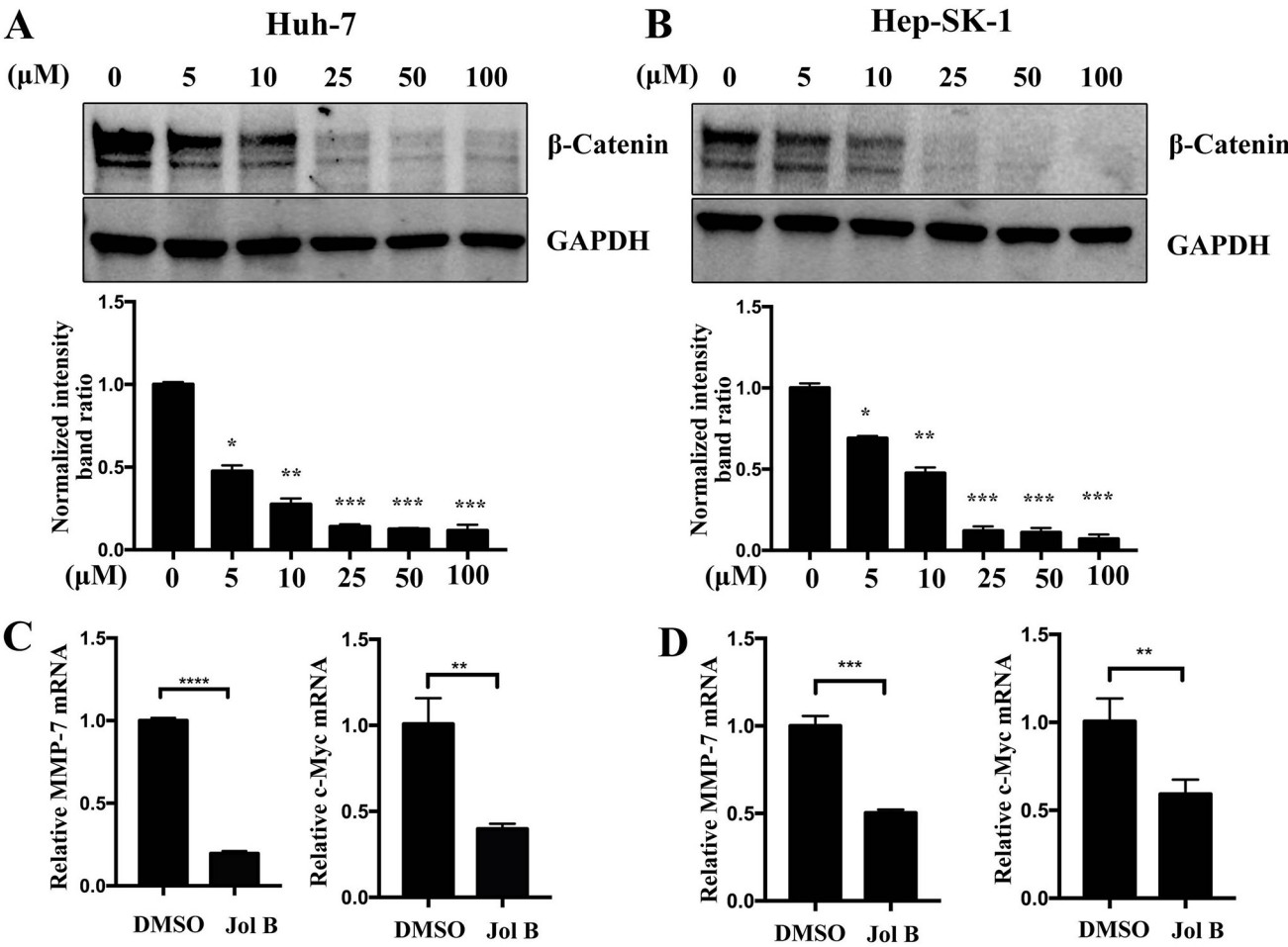

**Fig 3. The effects of Jolkinolide B on the β-catenin pathway in HCC cells.** (A) Huh-7 and (B) SK-Hep-1 cells were treated with the indicated concentrations of Jolkinolide B for 48 h. The β-catenin protein expression was analyzed using western blotting. GAPDH was used as an internal control. The relative protein intensities were analyzed. (C, D) Huh-7 and SK-Hep-1 cells were treated with 10 μM Jolkinolide B for 48 h. The mRNA expression of *MMP-7* and *C-MYC* in Huh-7 (C) and SK-Hep-1 (D) cells analyzed using RT-qPCR method. GAPDH was used as an internal control. *$p$ <0.05, **$p$ <0.01, ***$p$ <0.001 and ****$p$ <0.0001.

investigated the effect of MSI2 in Jolkinolide B-induced EMT. The results demonstrated that MSI2 reversed Jolkinolide B-induced upregulation of E-cadherin and downregulation of vimentin (Fig 6B). Finally, β-catenin protein expression in Huh-7 and SK-Hep-1 cells was also reversed by MSI2 overexpression in Jolkinolide B-induced downregulation of β-catenin signaling (Fig 6C).

## Discussion

Recent studies have shown that Jolkinolide B exerts anti-tumor activities against many tumors by regulating tumor metastasis and apoptosis through different mechanisms [13]. During tumor metastasis, epithelial cells lose polarity, which is acquired by mesenchymal cells, contributing to tumor cell migration and invasion [16–18]. In this study, we demonstrated that Jolkinolide B reduced the migration and invasion of Huh-7 and SK-Hep-1 cells. E-cadherin is a biomarker of epithelial cells, whereas vimentin is a biomarker of mesenchymal cells.

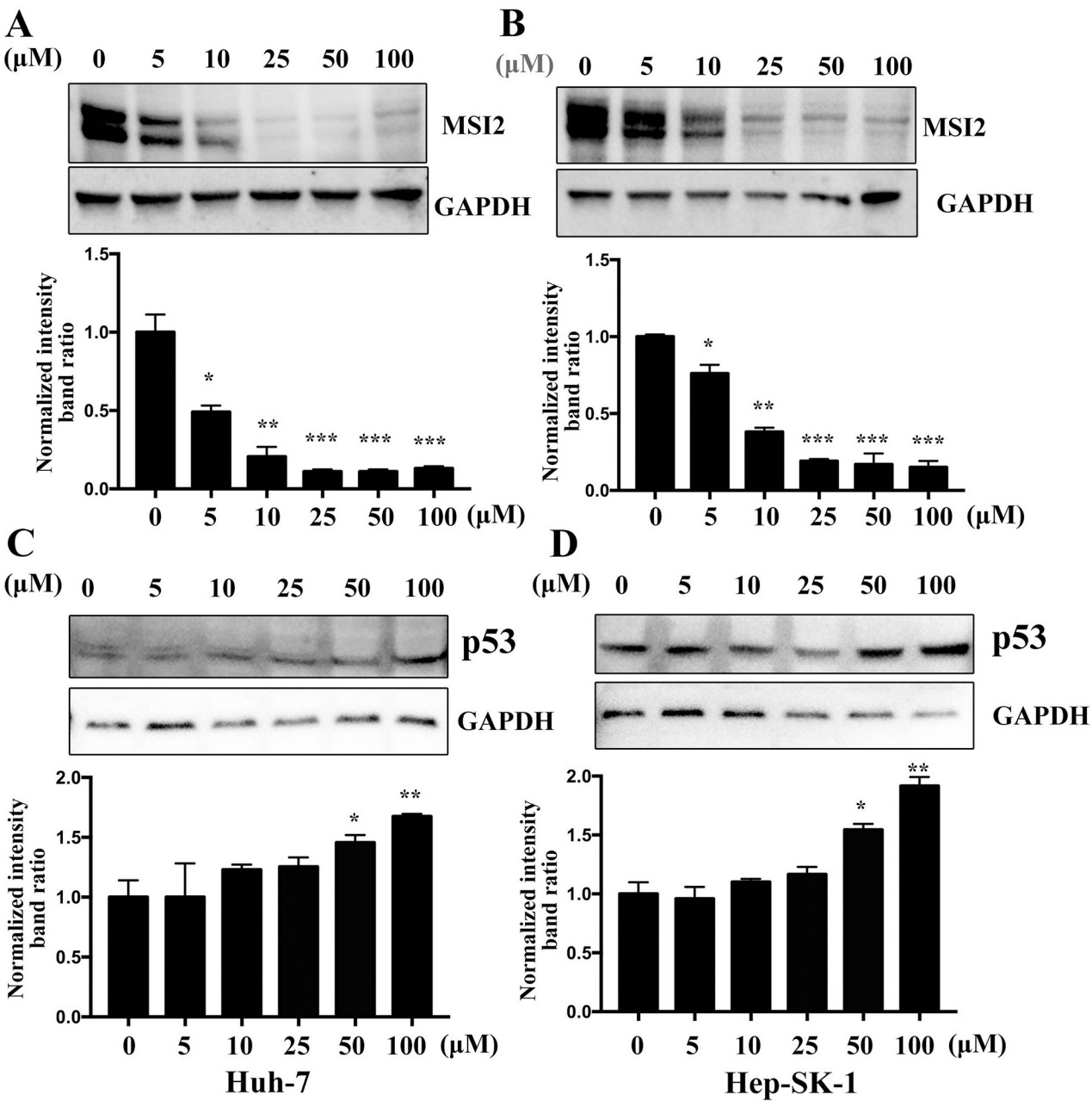

**Fig 4. Jolkinolide B downregulates the protein of MSI2 and the p53 expression in HCC cells.** (A, C) Huh-7 and (B, D) SK-Hep-1 cells were treated with the indicated concentrations of Jolkinolide B for 48 h. The MSI2 and p53 protein expressions were analyzed using western blotting. GAPDH was used as an internal control. *$p < 0.05$, **$p < 0.01$ and ***$p < 0.001$.

Therefore, EMT is characterized by the loss of E-cadherin and acquisition of vimentin [19–21]. In this study, Jolkinolide B inhibited EMT progression, resulting in an increase in E-cadherin and a decrease in vimentin levels. Cellular apoptosis is programmed cell death. The Bax/BCL-2 ratio changes play an important role in mediating the downstream apoptotic cascades that leads to apoptosis [22, 23]. Herein, we demonstrated through western blot analysis that

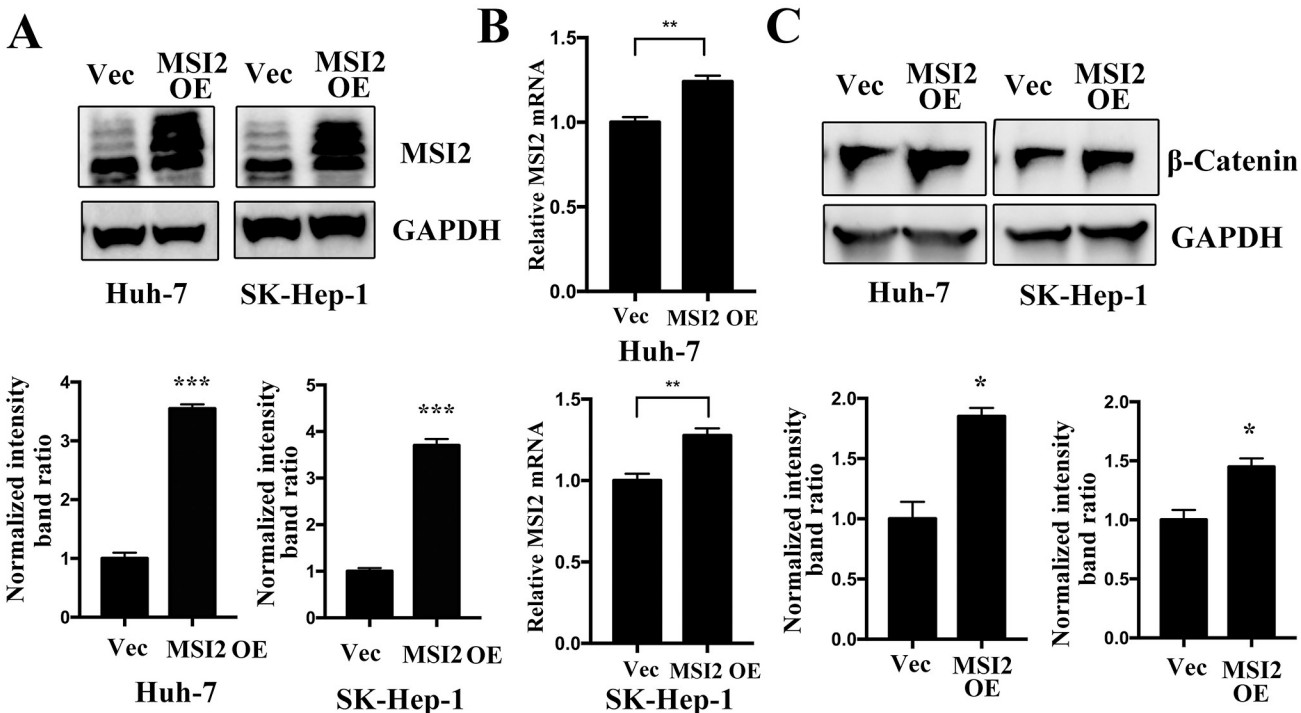

**Fig 5. MSI2 overexpression promotes β-catenin expression in HCC cells.** Huh-7 and SK-Hep-1 cells were transfected with lentivirus-mediated MSI2 overexpression plasmids for 48 h, the MSI2 protein expression (A) and MSI2 mRNA expression (B) were analyzed using western blotting and RT-qPCR methods, respectively. (C) β-catenin protein was detected in Huh-7 and SK-Hep-1 cells after MSI2 overexpression. The relative protein intensities were analyzed. $^*p < 0.05$, $^{**}p < 0.01$ and $^{***}p < 0.001$.

Jolkinolide B increased Bax protein levels and decreased BCL-2 protein expression. Our results showed that Jolkinolide B inhibits HCC cell migration and induces apoptosis.

Several studies showed that Wnt/β-catenin plays an important role in mediating tumor cell apoptosis and migration. Therefore, the inactivation of Wnt/β-catenin signaling pathway has been proposed as a potential therapeutic target [24–26]. For instance, paeonol exerts anti-colorectal cancer effects by inhibiting the Wnt/β-catenin signaling, whereas baicalin inhibits highly aggressive breast cancer by targeting Wnt/β-catenin signaling [27, 28]. In our study, we revealed that Jolkinolide B inhibits the β-catenin signaling by reducing the β-catenin protein expression.

MSI2, which is highly expressed in the hematopoietic system, contributes to hematopoietic stem cell regulation, and sperm and embryo formation [29, 30]. Recent studies have shown that MSI2 is closely related to tumor cell proliferation, invasion, migration, and apoptosis, and is an important regulator of various signaling pathways [6, 10, 31]. In addition, MSI2 is critical for HCC progression; for instance, MSI2 contributes to HCC cell stemness and chemoresistance, and MSI2 promotes hepatitis B virus-related HCC [7, 32]. Therefore, targeting MSI2 may be an effective approach for treating HCC. In this study, we used Jolkinolide B to target MSI2 and found that Jolkinolide B reduced MSI2 expression.

In summary, this study found that Jolkinolide B inhibited HCC by promoting apoptosis, reducing cell migration, and inducing β-catenin signaling inactivation. In addition, we further discovered that Jolkinolide B reduces the MSI2 expression, which can reverse Jolkinolide B-induced anti-tumor effects (Fig 6D). This study provides a novel therapeutic target for treating HCC.

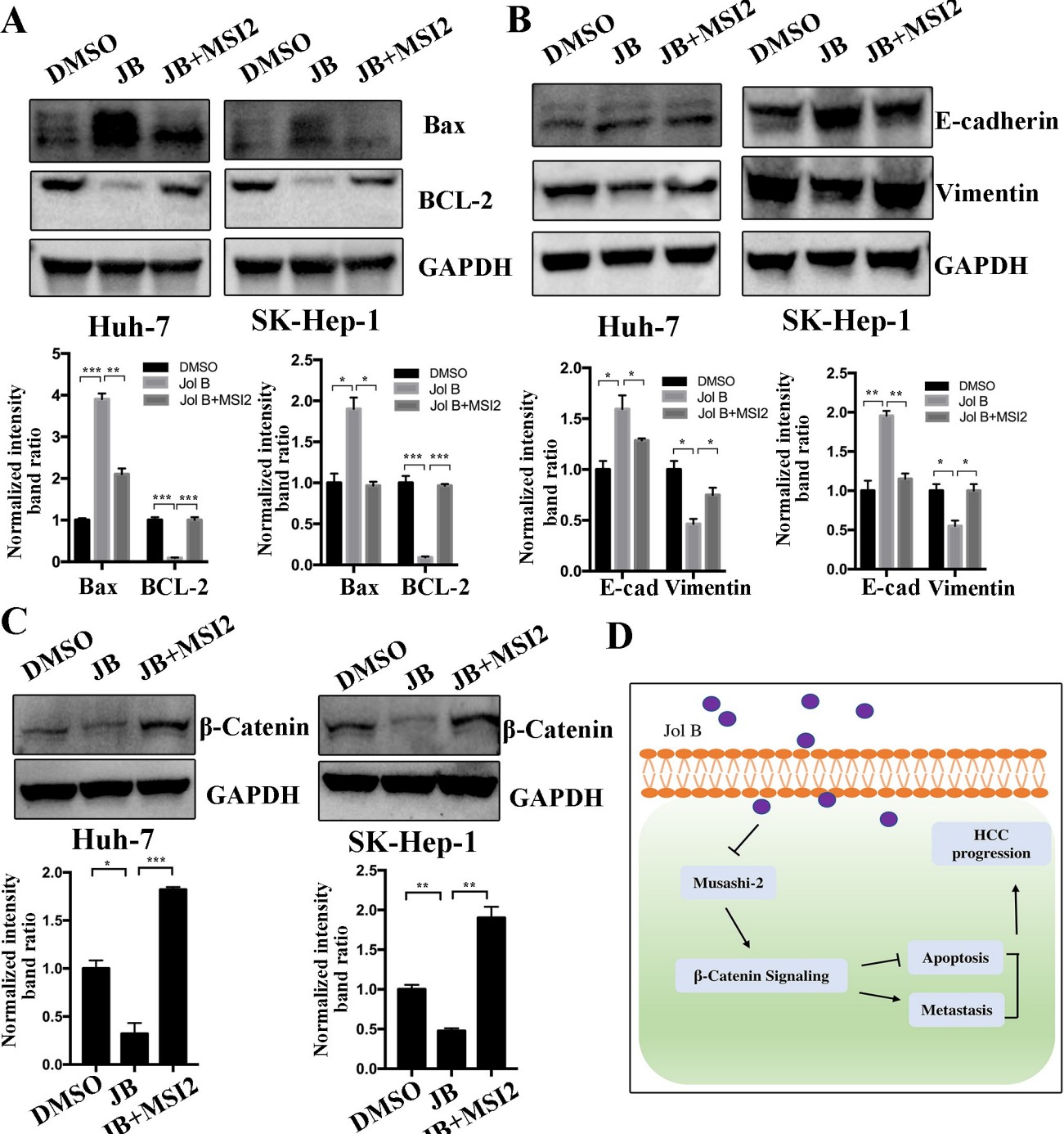

**Fig 6. Overexpression of MSI2 reverses Jolkinolide B-promoted apoptosis and Jolkinolide B-inhibited EMT and β-catenin signaling in HCC cells.**
Huh-7 and SK-Hep-1 cells were treated with DMSO, 10 μM Jolkinolide B, or Jolkinolide B together with MSI2 plasmids. Then, the protein expressions of (A) Bax, BCL-2, (B) E-cadherin, vimentin and (C) β-catenin were analyzed using western blotting. GAPDH was used as an internal control. The relative protein intensities were analyzed. (D) Proposed working model of Jolkinolide B-induced inhibition of HCC. $^{*}p < 0.05$, $^{**}p < 0.01$ and $^{***}p < 0.001$.

## Supporting information

**S1 File.** Slide 1: Original images for Fig 2A, 2B, 2E. Slide 2: Original immunoblots for Fig 2C, 2D. Slide 3: Original immunoblots for Fig 3. Slide 4: Original immunoblots for Fig 4. Slide 5: Original immunoblots for Fig 5A, 5C. Slide 6: Original immunoblots for Fig 6A, 6B. Slide 7: Original immunoblots for Fig 6C. Sheet 1: Original CCK8 of cell viability for Fig 1A-1C. Original RT-qPCR for Figs 3C, 3D, 5B.
(ZIP)

## Author Contributions

**Conceptualization:** Tianchun Wu, Han Yang.

**Formal analysis:** Tianchun Wu, Jinjin Li.

**Funding acquisition:** Tianchun Wu.

**Investigation:** Tianchun Wu, Liushun Feng.

**Methodology:** Tianchun Wu, Jie Li.

**Software:** Hongbo Fang.

**Visualization:** Xiaoyi Shi.

**Writing – original draft:** Tianchun Wu.

**Writing – review & editing:** Tianchun Wu.

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
