## [Decision Letter · Decision Letter 0]

11 Dec 2023

PONE-D-23-36751Jolkinolide B inhibits the progression of hepatocellular carcinoma through regulating musashi-2 proteinPLOS ONE

Dear Dr. Wu

Thank you for submitting your manuscript to PLOS ONE. After careful consideration, we feel that it has merit but does not fully meet PLOS ONE’s publication criteria as it currently stands. Therefore, we invite you to submit a revised version of the manuscript that addresses the points raised during the review process.

We look forward to receiving your revised manuscript.

Kind regards,

Kishor Pant

Academic Editor

PLOS ONE

Journal Requirements:

4. Thank you for stating the following financial disclosure: "Medical scientific and technological research joint construction projects of Henan province,2020. NO.LHGJ20200361."

7. PLOS requires an ORCID iD for the corresponding author in Editorial Manager on papers submitted after December 6th, 2016. Please ensure that you have an ORCID iD and that it is validated in Editorial Manager. To do this, go to ‘Update my Information’ (in the upper left-hand corner of the main menu), and click on the Fetch/Validate link next to the ORCID field. This will take you to the ORCID site and allow you to create a new iD or authenticate a pre-existing iD in Editorial Manager. Please see the following video for instructions on linking an ORCID iD to your Editorial Manager account: https://www.youtube.com/watch?v=_xcclfuvtxQ

Reviewers' comments:

Reviewer's Responses to Questions

**Comments to the Author**

1. Is the manuscript technically sound, and do the data support the conclusions?

Reviewer #1: Yes

Reviewer #2: No

2. Has the statistical analysis been performed appropriately and rigorously? 

Reviewer #1: Yes

Reviewer #2: Yes

3. Have the authors made all data underlying the findings in their manuscript fully available?

Reviewer #1: Yes

Reviewer #2: Yes

4. Is the manuscript presented in an intelligible fashion and written in standard English?

Reviewer #1: Yes

Reviewer #2: Yes

5. Review Comments to the Author

Reviewer #1: Based on the information presented in the study, the following concerns are proposed as prospective enhancements:

Major revision.

The manuscript represents the role of Jolkinolide B in the progression of hepatocellular carcinoma through the regulation of Musashi-2 protein; however, the authors did not investigate the mechanism in detail. They may detect changes in the survival signaling pathways such as STAT3, p53, AP-1, ATF6, or PI3K/AKT pathways.

Minor revision.

1. The manuscript is standard English, though some errors should be rectified.

2. In the legends of Figures 3 and 4, gapdh should be GAPDH, and also in materials and method of RT-qPCR.

3. Project No. in the funding section needs to be included.

Reviewer #2: The manuscript entitled “Jolkinolide B inhibits the progression of hepatocellular carcinoma through regulating musashi-2 protein” provides new insights that, Jolkinolide B as a potential candidate against hepatocellular carcinoma. It needs major revision to make the suitable for publication.

1. IC50 value of Jolkinolide B should be calculated.

2. Authors should confirm the apoptosis induced by Jolkinolide B, Annexin V or Tunnel assay in both lines.

3. In figure 2 legend, authors should mention concentration of JB and time of treatment of better understanding.

4. Similarly in Figure 3 and 4 legend, mention the treatment time.

5. Authors also should include the effect of JB on normal hepto cell lines.

6. All Western blots in manuscript should be quantified.

6. PLOS authors have the option to publish the peer review history of their article (what does this mean?). If published, this will include your full peer review and any attached files.

Reviewer #1: No

Reviewer #2: No

---

## [Author Response · Author response to Decision Letter 0]

3 Feb 2024

Dear Editor and Reviewer:

Thank you for your letter and for the reviewer’s comments concerning our manuscript entitled “Jolkinolide B inhibits the progression of hepatocellular carcinoma by regulating Musashi-2 protein”. We very much appreciate all of the comments received in the review. Those comments are all valuable and very helpful for revising and improving our paper, as well as the important guiding significance to our researchers. We have studied these comments carefully and have made corrections accordingly. Our responses are given in a point-by-point manner below. Changes to the manuscript are shown with track changes.

We hope the revised version is now suitable for publication and look forward to hearing from you in due course.

Responds to the reviewer’s comments:

Reviewer #1: Based on the information presented in the study, the following concerns are proposed as prospective enhancements:

Major revision.

The manuscript represents the role of Jolkinolide B in the progression of hepatocellular carcinoma through the regulation of Musashi-2 protein; however, the authors did not investigate the mechanism in detail. They may detect changes in the survival signaling pathways such as STAT3, p53, AP-1, ATF6, or PI3K/AKT pathways.

Reply:

We thank the reviewer’s question and following the reviewer’s suggestion, as p53 protein having an important role in inhibiting cancer progression, we have now detected the p53 protein expression in Jolkinolide B-induced HCC cell lines in Supplementary information.

Minor revision.

The manuscript is standard English, though some errors should be rectified.

Reply:

We thank the reviewer’s question. We have now reviewed and revised our manuscript with the help of native English-speakers.

In the legends of Figures 3 and 4, gapdh should be GAPDH, and also in materials and method of RT-qPCR.

Reply:

We thank the reviewer’s question. We have now revised the writings in the manuscript.

3. Project No. in the funding section needs to be included.

Reply:

We thank the reviewer’s question. The project number in the funding section has been included in the revised manuscript.

Reviewer #2: The manuscript entitled “Jolkinolide B inhibits the progression of hepatocellular carcinoma through regulating musashi-2 protein” provides new insights that, Jolkinolide B as a potential candidate against hepatocellular carcinoma. It needs major revision to make the suitable for publication.

IC50 value of Jolkinolide B should be calculated.

Reply:

We thank the reviewer’s question and the IC50 values of Jolkinolide B in HCC cell lines have been calculated.

2.Authors should confirm the apoptosis induced by Jolkinolide B, Annexin V or Tunnel assay in both lines.

Reply:

We thank the reviewer’s question. We have detected the apoptosis of HCC cell lines treated by DMSO or Jolkinolide B administration by flow cytometry. 

3. In figure 2 legend, authors should mention concentration of JB and time of treatment of better understanding.

Reply:

We thank the reviewer’s question and we are sorry for having not described the experiments more clearly. We have now provided the description of Jolkinolide B concentration and time of treatment in the revised manuscript.

4. Similarly in Figure 3 and 4 legends, mention the treatment time.

Reply:

We thank the reviewer’s question and we are sorry for having not described the experiments more clearly. We have now provided more precise description of Jolkinolide B’s treatment time in the revised manuscript.

5. Authors also should include the effect of JB on normal hepto cell lines.

Reply:

We thank the reviewer’s question and we now have detected the effect of Jolkinolide B on normal hepto cell line.

6. All Western blots in manuscript should be quantified.

Reply:

We thank the reviewer’s question and following the reviewer’s suggestion, we have quantified the normalized intensity band ratios of all western blots in revised manuscript.

---

## [Decision Letter · Decision Letter 1]

19 Feb 2024

Jolkinolide B inhibits the progression of hepatocellular carcinoma by regulating Musashi-2 protein

PONE-D-23-36751R1

Dear Dr. Wu,

We’re pleased to inform you that your manuscript has been judged scientifically suitable for publication and will be formally accepted for publication once it meets all outstanding technical requirements.

Kind regards,

Kishor Pant

Academic Editor

PLOS ONE

Reviewers' comments:

Reviewer's Responses to Questions

**Comments to the Author**

1. If the authors have adequately addressed your comments raised in a previous round of review and you feel that this manuscript is now acceptable for publication, you may indicate that here to bypass the “Comments to the Author” section, enter your conflict of interest statement in the “Confidential to Editor” section, and submit your "Accept" recommendation.

Reviewer #1: All comments have been addressed

Reviewer #2: (No Response)

2. Is the manuscript technically sound, and do the data support the conclusions?

Reviewer #1: Yes

Reviewer #2: Yes

3. Has the statistical analysis been performed appropriately and rigorously? 

Reviewer #1: Yes

Reviewer #2: Yes

4. Have the authors made all data underlying the findings in their manuscript fully available?

Reviewer #1: Yes

Reviewer #2: (No Response)

5. Is the manuscript presented in an intelligible fashion and written in standard English?

Reviewer #1: Yes

Reviewer #2: Yes

6. Review Comments to the Author

Reviewer #1: (No Response)

Reviewer #2: Manuscript looks good after revision and authors have made necessary changes according to reviewer's comments. So I would recommend for publication.

7. PLOS authors have the option to publish the peer review history of their article (what does this mean?). If published, this will include your full peer review and any attached files.

Reviewer #1: No

Reviewer #2: No

---

## [Editor Report · Acceptance letter]

5 Apr 2024

PONE-D-23-36751R1 

PLOS ONE

Dear Dr. Wu, 

I'm pleased to inform you that your manuscript has been deemed suitable for publication in PLOS ONE. Congratulations! Your manuscript is now being handed over to our production team.

Kind regards, 

on behalf of

Dr. Kishor Pant 

Academic Editor

PLOS ONE